# *In Vitro* Activities and Inoculum Effects of Cefiderocol and Aztreonam-Avibactam against Metallo-$\beta$-Lactamase-Producing *Enterobacteriaceae*

Yu-Shan Huang,[a,b] Pao-Yu Chen,[a,b] Pei-Chun Chou,[c] (ORCID) Jann-Tay Wang[a]

[a]Department of Internal Medicine, National Taiwan University Hospital, Taipei, Taiwan
[b]Graduate Institute of Clinical Medicine, National Taiwan University College of Medicine, Taipei, Taiwan
[c]Laboratory of Infectious Disease, Department of Internal Medicine, National Taiwan University Hospital, Taipei, Taiwan

**ABSTRACT** Cefiderocol and aztreonam-avibactam (ATM-AVI) both had activity against carbapenem-resistant Gram-negative bacilli, including those that produce metallo-$\beta$-lactamases (MBLs). We compared the *in vitro* activities and inoculum effects of these antibiotics against carbapenemase-producing *Enterobacteriaceae* (CPE), especially MBL-producing isolates. The MICs of cefiderocol and ATM-AVI were determined using broth microdilution method for a 2016 to 2021 collection of *Enterobacteriaceae* isolates which produced MBL, KPC, or OXA-48-like carbapenemases. MICs with high bacteria inoculum were also evaluated for susceptible isolates. A total of 195 CPE were tested, including 143 MBL- (74 NDM, 42 IMP, and 27 VIM), 38 KPC-, and 14 OXA-48-like-producing isolates. The susceptible rates of MBL-, KPC-, and OXA-48-like producers to cefiderocol were 86.0%, 92.1%, and 92.9%, respectively, and that to ATM-AVI were 95.8%, 100%, and 100%, respectively. NDM producers displayed lower susceptibility and higher $MIC_{50}s/MIC_{90}s$ of cefiderocol (78.4%, 2/16 mg/L) than IMP (92.9%, 0.375/4 mg/L) and VIM (96.3%, 1/4 mg/L) producers. NDM- and VIM-producing *Escherichia coli* showed lower susceptibility to ATM-AVI (77.3% and 75.0%, respectively) compared to MBL-CPE of other species (100% susceptible). Inoculum effects for cefiderocol and ATM-AVI were observed among 95.9% and 95.2% of susceptible CPE, respectively. A switch from susceptible to resistant category was observed in 83.6% (143/171) of isolates for cefiderocol and 94.7% (179/189) for ATM-AVI. Our results revealed that NDM-producing *Enterobacteriaceae* had lower susceptibility to cefiderocol and ATM-AVI. Prominent inoculum effects on both antibiotics were observed for CPE, which suggested a risk of microbiological failure when they were used for CPE infections with high bacteria burden.

**IMPORTANCE** The prevalence of infections caused by carbapenem-resistant *Enterobacteriaceae* is increasing worldwide. Currently, therapeutic options for metallo-$\beta$-lactamase (MBL)-producing *Enterobacteriaceae* remain limited. We demonstrated that clinical metallo-$\beta$-lactamase (MBL)-producing *Enterobacteriaceae* isolates were highly susceptible to cefiderocol (86.0%) and aztreonam-avibactam (ATM-AVI) (95.8%). However, inoculum effects on cefiderocol and ATM-AVI were observed for over 90% of susceptible carbapenemase-producing *Enterobacteriaceae* (CPE) isolates. Our findings highlight a potential risk of microbiological failure when using monotherapy with cefiderocol or ATM-AVI to treat severe CPE infection.

**KEYWORDS** carbapenem-resistant *Enterobacteriaceae*, metallo-$\beta$-lactamase, cefiderocol, $\beta$-lactam and $\beta$-lactamase inhibitor, inoculum effect

**Ad Hoc Peer Reviewer** (ORCID) Dennis Nurjadi, Universitat zu Lubeck

Address correspondence to Jann-Tay Wang, wang.jt1968@gmail.com.

The authors declare no conflict of interest.

The prevalence of carbapenem resistance has increased among clinical isolates of *Enterobacteriaceae* in the past decade. *Enterobacteriaceae* carrying carbapenemase-encoding plasmids raised special concern because the antimicrobial resistance could be

transferred between different strains and species (1). The predominant carbapenemase types in *Enterobacteriaceae* varied across geographic regions. In Southeast Asia, metallo-$\beta$-lactamases (MBLs) and oxacillinase-48 (OXA-48) were the main carbapenemases in *Enterobacteriaceae*, whereas *Klebsiella pneumoniae* carbapenemase (KPC) was the most common carbapenemase in North America (2). In Europe, OXA-48 and KPC were more frequently detected than MBLs, but an increase in NDM-1 producers was noted since 2022, which was associated with the migration of people (3, 4).

Newer $\beta$-lactam with $\beta$-lactamase inhibitor combinations that were actively against carbapenem-resistant *Enterobacteriaceae* (CRE) had been developed in recent years, but MBLs could hydrolyze most of these antibiotics, such as ceftazidime-avibactam, ceftolozane-tazobactam, and meropenem-vaborbactam (5). Therefore, treatment options for MBL carbapenemase-producing *Enterobacteriaceae* (CPE) remained limited, and 14-day mortality-following infections with these pathogens could be up to 60% (6). Among newer antibiotics, a siderophore cephalosporin cefiderocol and aztreonam combined with avibactam (ATM-AVI) were proposed for treating infections with MBL CPE based on the *in vitro* susceptibility data and clinical reports (7). Nonetheless, MBL CPE were not universally susceptible to cefiderocol or ATM-AVI. Published studies reported higher cefiderocol minimum inhibitory concentrations (MICs) for New Delhi metallo-$\beta$-lactamase (NDM)-producing *Enterobacteriaceae* than those carrying other MBLs (8, 9). For ATM-AVI, Vasoo et al. found that two (1.2%) of 172 CPE had elevated MICs of ATM-AVI (8/4 and 16/4 mg/L), and both produced NDM carbapenemase (10). Data on the susceptibilities of multidrug-resistant *Enterobacteriaceae* to cefideocol or ATM-AVI have been increasingly reported (11, 12). However, many did not compare the results across bacterial species or types of MBL.

The MIC of many beta-lactam antibiotics can be influenced by the inoculum effect. Inoculum effect refers to a laboratory phenomenon in which the MIC of an antibiotic increases significantly when the number of inoculated organisms is increased. *In vitro* inoculum effect of cefazolin has been associated with worse outcomes when used to treat methicillin-susceptible *Staphylococcus aureus* bacteremia (13). In the literature, cephalosporins displayed inoculum effect against various bacterial pathogens, including *Escherichia coli* and *Klebsiella pneumoniae* (14). For extended-spectrum $\beta$-lactamases-producing *Enterobacteriaceae*, both cephalosporins and aztreonam tended to show inoculum effect (14, 15). At present, only three studies investigated the inoculum effect of cefiderocol or ATM-AVI against CRE (16–18), and the numbers of enrolled MBL CPE isolates were small. In addition, the impact of carbapenemase types or species of *Enterobacteriaceae* on inoculum effect had rarely been evaluated.

In this study, we aimed to compare the *in vitro* activities of cefiderocol and ATM-AVI against different types of CPE, especially the MBL-producing isolates, and determine the MICs at standard and high inoculum.

## RESULTS

A total of 195 CPE were tested, including 143 MBL- (74 NDM, 42 imipenemase [IMP], and 27 verona integron-encoded [VIM] type carbapenemase), 38 KPC-, and 14 OXA-48-like-producing *Enterobacteriaceae* isolates. Of the 74 NDM producers, 49 (66.2%) were NDM-1, 24 (32.4%) were NDM-5, and one (1.4%) was NDM-4. All IMP producers were IMP-8, and all VIM producers were VIM-1. The subtypes of KPC carbapenemase included KPC-2 (89.5%, 34/38), KPC-17 (5.3%, 2/38), KPC-3 (2.6%, 1/38), and KPC-65 (2.6%, 1/38). Of the 14 OXA-48-like CPE, 13 were OXA-48 and one was OXA-181.

Among the 195 CPE isolates, *K. pneumoniae* ($n = 79$, 40.5%) was the most common species, followed by *E. coli* ($n = 56$, 28.7%) and *E. cloacae* complex ($n = 44$, 22.6%). Other species included nine *Citrobacter freundii*, six *Klebsiella oxytoca*, and one *Klebsiella aerogenes*. The majority of CPE were clinical isolates from urine ($n = 68$, 34.9%), blood ($n = 50$, 25.6%), sputum ($n = 37$, 19.0%), skin pus or wound ($n = 14$, 7.2%), body fluid ($n = 5$, 2.6%), and catheter tip ($n = 1$, 0.5%). Twenty (10.3%) of the 195 CPE were from anal screening cultures.

**TABLE 1** *In vitro* susceptibilities and MICs (mg/L) of cefiderocol and aztreonam-avibactam against carbapenemase-producing *Enterobacteriaceae*

| Carbapenemase type and bacteria species | AMK[c] S (%) | ATM S (%) | CIP S (%) | COL I (%) | TGC S (%) | Cefiderocol | | | | Aztreonam-avibactam[a] | | | |
|---|---|---|---|---|---|---|---|---|---|---|---|---|---|
| | | | | | | MIC range | $MIC_{50}$ | $MIC_{90}$ | S (%) | MIC range | $MIC_{50}$ | $MIC_{90}$ | S (%) |
| MBL (*n* = 143) | 92.3 | 26.8 | 30.8 | 92.3 | 98.6 | ≤0.06–>32 | 2 | 8 | 86.0 | ≤0.06–16 | 0.25 | 1 | 95.8 |
| | | | | | | | | | | | | | |
| NDM (*n* = 74) | 90.5 | 18.9 | 12.2 | 93.2 | 100 | 0.25–>32 | 2 | 16 | 78.4 | ≤0.06–16 | 0.25 | 4 | 93.2 |
| *E. coli* (*n* = 22) | 95.5 | 18.2 | 4.5 | 100 | 100 | 0.5–16 | 4 | 16 | 72.7 | ≤0.06–16 | 2 | 8 | 77.3 |
| ECC (*n* = 31) | 100 | 3.2 | 6.5 | 100 | 100 | 1–>32 | 2 | 16 | 77.4 | ≤0.06–1 | 0.25 | 0.5 | 100 |
| *K. pneumoniae* (*n* = 14) | 78.6 | 42.9 | 35.7 | 85.7 | 100 | 0.5–8 | 1.5 | 4 | 92.9 | ≤0.06–1 | 0.188 | 0.5 | 100 |
| Other species[b] (*n* = 7) | 57.1 | 42.9 | 14.3 | 57.1 | 100 | 0.25–32 | 4 | 32 | 71.4 | ≤0.06–0.5 | 0.25 | 0.5 | 100 |
| | | | | | | | | | | | | | |
| IMP (*n* = 42) | 95.2 | 51.2 | 52.4 | 92.9 | 95.2 | ≤0.06–32 | 0.375 | 4 | 92.9 | ≤0.06–0.5 | 0.125 | 0.5 | 100 |
| *E. coli* (*n* = 17) | 94.1 | 62.5 | 52.9 | 100 | 100 | ≤0.06–4 | 0.25 | 4 | 100 | ≤0.06–0.5 | 0.125 | 0.5 | 100 |
| ECC (*n* = 13) | 92.3 | 30.8 | 53.8 | 76.9 | 92.3 | ≤0.06–32 | 1 | 16 | 76.9 | ≤0.06–0.5 | 0.125 | 0.5 | 100 |
| *K. pneumoniae* (*n* = 5) | 100 | 100 | 100 | 100 | 80.0 | ≤0.06–0.25 | 0.06 | 0.25 | 100 | ≤0.06–0.25 | 0.125 | 0.25 | 100 |
| Other species[b] (*n* = 7) | 100 | 28.6 | 42.9 | 100 | 100 | ≤0.06–4 | 0.5 | 4 | 100 | ≤0.06–0.5 | 0.125 | 0.5 | 100 |
| | | | | | | | | | | | | | |
| VIM (*n* = 27) | 92.6 | 11.1 | 48.1 | 88.9 | 100 | ≤0.06–8 | 1 | 4 | 96.3 | ≤0.06–16 | 0.25 | 0.25 | 96.3 |
| *E. coli* (*n* = 4) | 100 | 25.0 | 50.0 | 100 | 100 | ≤0.06–2 | 0.313 | 2 | 100 | ≤0.06–16 | 0.188 | 16 | 75.0 |
| *K. pneumoniae* (*n* = 21) | 90.5 | 9.5 | 47.6 | 85.7 | 100 | ≤0.06–8 | 1 | 4 | 95.2 | ≤0.06–0.5 | 0.25 | 0.25 | 100 |
| Other species[b] (*n* = 2) | 100 | 0.0 | 50.0 | 100 | 100 | 0.5–2 | 1.25 | 2 | 100 | ≤0.06–0.25 | 0.155 | 0.25 | 100 |
| | | | | | | | | | | | | | |
| KPC (*n* = 38) | 94.7 | 2.6 | 5.3 | 71.1 | 100 | ≤0.06–16 | 0.75 | 4 | 92.1 | ≤0.06–4 | 0.5 | 1 | 100 |
| OXA-48-like (*n* = 14) | 85.7 | 21.4 | 7.1 | 85.7 | 100 | ≤0.06–16 | 0.75 | 4 | 92.9 | ≤0.06–2 | 0.25 | 1 | 100 |
| Total (*n* = 195) | 92.3 | 21.6 | 24.1 | 87.7 | 99.0 | ≤0.06–>32 | 1 | 8 | 87.7 | ≤0.06–16 | 0.25 | 1 | 96.9 |

[a]Avibactam was tested at a fixed concentration of 4 mg/L.
[b]Other species included nine *Citrobacter freundii*, six *Klebsiella oxytoca*, and one *Klebsiella aerogenes*.
[c]AMK, amikacin; ATM, aztreonam; CIP, ciprofloxacin; COL, colistin; ECC, *Enterobacter cloacae* complex; I, intermediate; MBL, metallo-$\beta$-lactamase; MIC, minimum inhibitory concentration; S, susceptibility; TGC, tigecycline.

**Susceptibilities and MICs of CPE to antimicrobial agents.** Table 1 presents the MIC values and antimicrobial susceptibilities of the CPE isolates. The susceptibility rates of all CPE to amikacin, aztreonam, ciprofloxacin, and tigecycline were 92.3% (180/195), 21.5% (42/195), 24.1% (47/195), and 99.0% (193/195), respectively. A total of 87.7% (171/195) of CPE isolates had a colistin MIC ≤2 mg/L. The susceptible rates of MBL-, KPC-, and OXA-48-like CPE to cefiderocol were 86.0% (123/143), 92.1% (35/38) and 92.9% (13/14), respectively. Among MBL CPE, NDM producers showed lower susceptibility and higher $MIC_{50}$/$MIC_{90}$ values of cefiderocol (78.4% [58/74], 2/16 mg/L) than IMP (92.9% [39/42], 0.375/4 mg/L) and VIM (96.3% [26/27], 1/4 mg/L) producers. Fig. 1A shows the cumulative percentages of different CPE inhibited at various cefiderocol concentrations.

Among 24 cefiderocol nonsusceptible CPE, 16 (66.7%) carried NDM, of which 13 carried NDM-1 (seven *E. cloacae* complex, three *E. coli*, one *K. pneumoniae*, one *Klebsiella oxytoca*, and one *Citrobacter freundii*) and three carried NDM-5 (all were *E. coli*). The rest of eight cefiderocol nonsusceptible CPE carried IMP (*n* = 3), KPC (*n* = 3), VIM (*n* = 1), and OXA-181 (*n* = 1). Fig. S1A demonstrates cefiderocol MIC distributions for different species of MBL CPE. Overall, MBL-producing *E. cloacae* complex were less susceptible to cefiderocol than MBL-producing *E. coli* or *K. pneumoniae* isolates (77.3% [34/44] versus 86.0% [37/43] and 95.0% [38/40]).

ATM-AVI inhibited 95.8% (137/143) of MBL CPE and 100% of KPC CPE and OXA-48-like CPE at the 4/4 mg/L breakpoint. Although KPC CPE was 100% susceptible to ATM-AVI, it demonstrated higher $MIC_{50}$ of ATM-AVI (0.5 mg/L) than MBL CPE (0.25 mg/L) and OXA-48-like CPE (0.25 mg/L) (Table 1; Fig. 1B). The MIC distributions of ATM-AVI by species of MBL CPE are presented in Fig. S1B. NDM- and VIM-positive *E. coli* showed elevated $MIC_{90}$s (8 mg/L and 16 mg/L, respectively) and lower susceptibility to ATM-AVI (77.3% [17/22] and 75.0% [3/4], respectively) compared to MBL CPE of other species. All six ATM-AVI nonsusceptible CPE were *E. coli* (four NDM-5, one NDM-1, and one VIM-1). Among 24 cefiderocol nonsusceptible CPE isolates, 22 (91.7%) were inhibited by ATM-AVI at the 4/4 mg/L breakpoint. Two *E. coli* isolates demonstrated high MICs for cefiderocol and ATM-AVI, and both carried NDM carbapenemase (1 NDM-1 and 1 NDM-5).

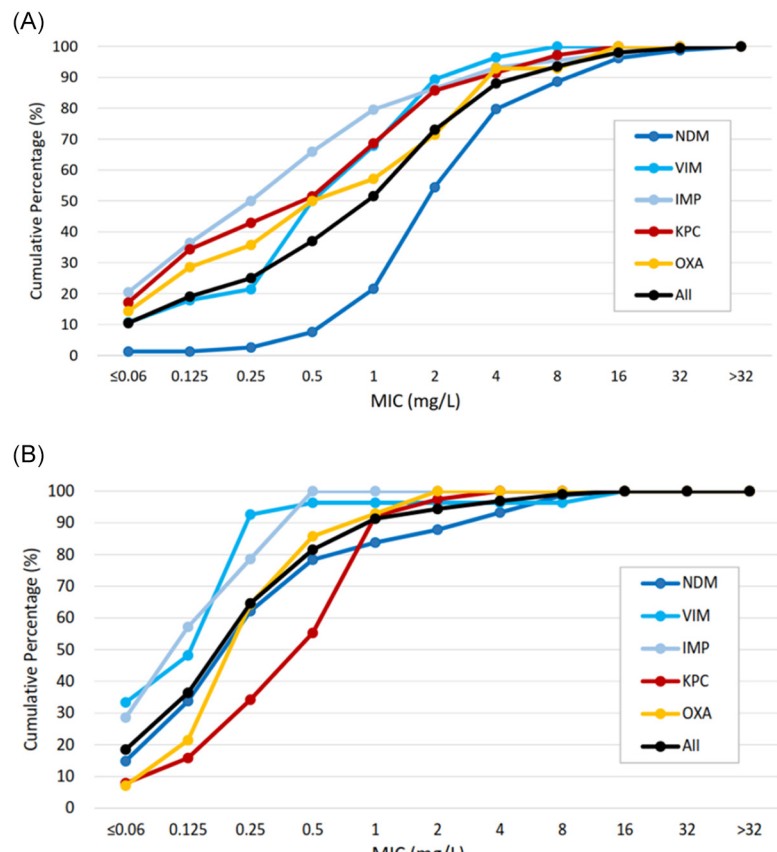

**FIG 1** *In vitro* activity of (A) cefiderocol and (B) aztreonam-avibactam against various carbapenemase-producing *Enterobacteriaceae*. Avibactam was tested at a fixed concentration of 4 mg/L.

**Clonal relationship of cefiderocol or ATM-AVI nonsusceptible isolates.** *E. coli* and *E. cloacae complex* accounted for the majority of cefiderocol or ATM-AVI nonsusceptible isolates, and NDM was the dominant carbapenemase type. Therefore, we investigated the clonal and epidemiological relatedness of 11 *E. coli* which were nonsusceptible to cefiderocol ($n = 5$), ATM-AVI ($n = 4$), or both ($n = 2$), as well as the 10 cefiderocol nonsusceptible *E. cloacae* complex. Among cefiderocol nonsusceptible *E. coli*, only two NDM-1-positive *E. coli* belonging to ST410 displayed the same pulsotype. As for ATM-AVI nonsusceptible *E. coli*, a cluster of four NDM-5-positive *E. coli* were found, and all belonged to ST617 (Fig. 2A). Two of the four ST617 *E. coli* isolates (M119 and M137) were from patients who had recently been discharged from two adjacent hospital wards within a 2-month period (Table S1). Among 10 cefiderocol non-susceptible *E. cloacae* complex (seven NDM-1 and three IMP-8 producers), five NDM-1-positive isolates with the same pulsotype were assigned ST316, while others showed distinct pulso-type and diverse ST (25, 66, 90, 1385) (Fig. 2B). Analysis of the epidemiologic data showed that three ST316 *E. cloacae* complex isolates (M89, M101, M124) were from patients who had been admitted to the hematology ward within the same year, but their admission periods did not overlap (Table S1).

**Inoculum effect.** Overall, inoculum effects for cefiderocol and ATM-AVI were detected in 95.9% (164/171) and 95.2% (180/189) of susceptible CPE isolates, respectively (Table 2). A switch from susceptible to resistant category was observed in 83.6% (143/171) of isolates for cefiderocol and 94.7% (179/189) for ATM-AVI. The inoculum effect of cefiderocol and ATM-AVI were observed in 98.4% (121/123) and 94.2% (129/137) of MBL CPE isolates, respectively. Compared to MBL CPE, KPC CPE showed a significantly lower rate of inoculum effect for cefiderocol (98.4% [121/123] versus 88.6% [31/35], $P = 0.022$). The frequency of inoculum effect for cefiderocol or ATM-AVI were comparable among NDM, IMP, and VIM CPE. Over 90% of carbapenemase-producing *E. coli*, *E. cloacae* complex, and *K. pneumoniae*

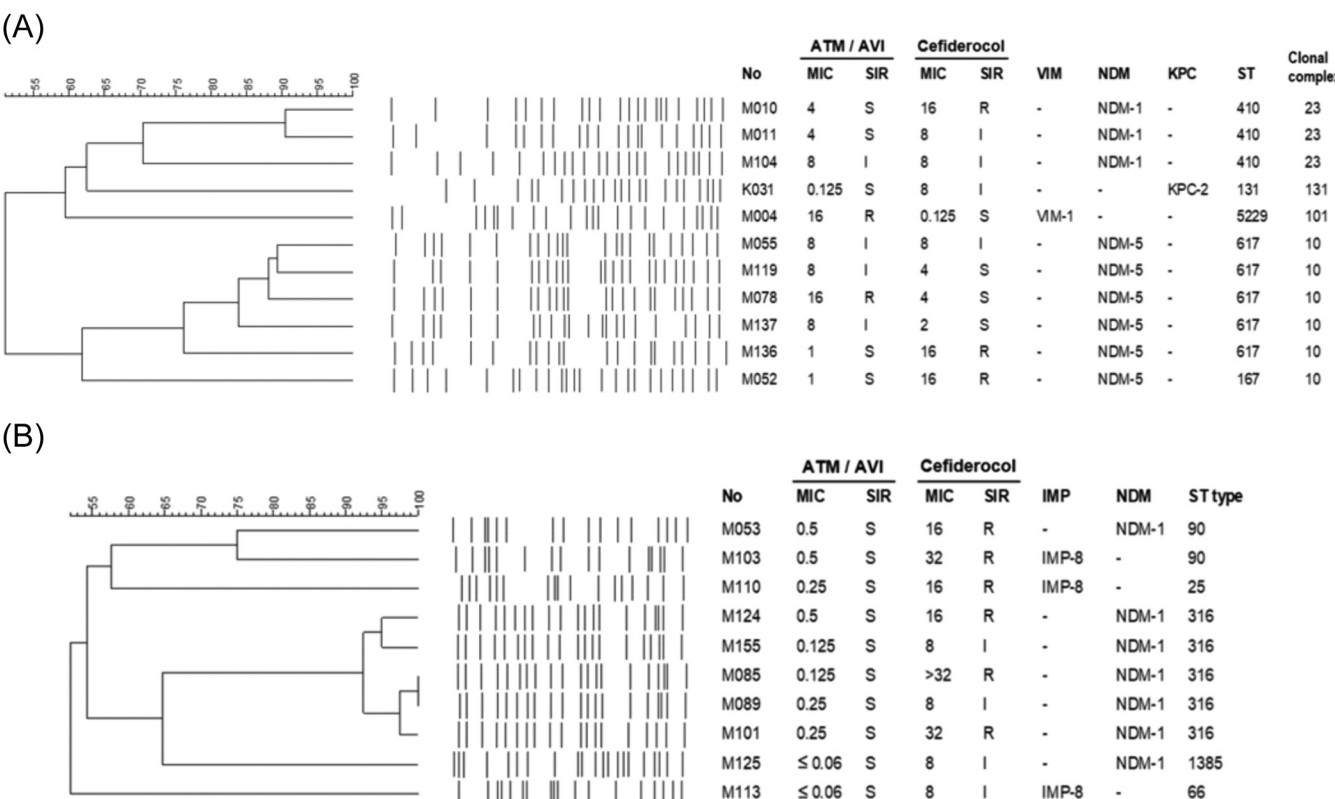

**FIG 2** Comparisons of (A) aztreonam-avibactam or cefedirocol nonsusceptible *E. coli* and (B) cefedirocol nonsusceptible *Enterobacter cloacae* complex isolates by PFGE dendrogram and MLST. Avibactam was tested at a fixed concentration of 4 mg/L.

isolates displayed an inoculum effect for cefiderocol, whereas for ATM-AVI, inoculum effect was less commonly detected in carbapenemase-producing *E. coli* (82.0%, 41/50) than in carbapenemase-producing *E. cloacae* complex (100%, 44/44) or *K. pneumoniae* (100%, 79/79) (Table 2).

## DISCUSSION

This *in vitro* study demonstrated high susceptibility of MBL CPE to cefiderocol (86.0%) and ATM-AVI (95.8%). CPE with NDM carbapenemase showed higher MICs of cefiderocol and ATM-AVI than those with VIM or IMP. By species, MBL-producing *E. cloacae* complex were less susceptible to cefiderocol than *E. coli* or *K. pneumoniae*, whereas MBL-producing *E. coli* were more frequently associated with elevated ATM-AVI MICs compared to other species. Over 90% of CPE isolates that were susceptible to cefiderocol or ATM-AVI at standard inoculum exhibited 8-fold or greater increase in MICs at high inoculum.

MBL-producing *Enterobacteriaceae* were more susceptible to ATM-AVI than cefiderocol in our study. Likewise, a higher susceptibility to ATM-AVI (70.3%) than cefiderocol (45.3%, using the CLSI breakpoint) among MBL CPE was reported by Terrier et al. (19). The detailed $\beta$-lactamases profile of MBL producers in Terrier's work was not provided, but in another study published by the same group, presence of extended-spectrum $\beta$-lactamases such as BEL and SHV conferred a greater impact on cefiderocol MICs than ATM-AVI MICs among *E. coli* (20). Furthermore, Kocer et al. showed that the acquisition of extrachromosomal fec operon in conjunction with VIM-type MBL could lead to a significant elevation of the cefiderocol MIC (21). These results indicated that the presence of transferable resistance determinant or other $\beta$-lactamases among MBL CPE might affect their susceptibilities to ATM-AVI or cefiderocol.

Isolates harboring NDM carbapenemases demonstrated higher cefiderocol MICs than other CPE. This finding was in line with previous reports (22, 23). Mechanisms responsible for cefiderocol resistance among NDM CPE included increased expression of $bla_{\mathrm{NDM}}$ gene

**TABLE 2** The frequency of inoculum effect among susceptible carbapenemase-producing *Enterobacteriaceae* isolates[a,b]

| Antimicrobial Agent | Total % (n/n) | Metallo-β-Lactamase, % (n/n) | | | | KPC, % (n/n) | OXA-48-like, % (n/n) | P-value | | |
| | | All MBL | NDM | IMP | VIM | | | MBL vs. KPC vs. OXA-48-like | MBL vs. KPC | MBL vs. OXA-48-like |
|---|---|---|---|---|---|---|---|---|---|---|
| Cefiderocol | 95.9 (164/171) | 98.4 (121/123) | 100 (58/58) | 94.5 (37/39) | 100 (26/26) | 88.6 (31/35) | 92.3 (12/13) | 0.027 | 0.022 | 0.262 |
| E. coli (n = 49) | 93.9 (46/49) | 94.6 (35/37) | 100 (16/16) | 88.2 (15/17) | 100 (4/4) | 100 (7/7) | 80.0 (4/5) | | | |
| ECC (n = 34) | 100 (34/34) | 100 (34/34) | 100 (24/24) | 100 (10/10) | NA | NA | NA | | | |
| K. pneumoniae (n = 74) | 94.6 (70/74) | 100 (38/38) | 100 (13/13) | 100 (5/5) | 100 (20/20) | 85.7 (24/28) | 100 (8/8) | | | |
| Aztreonam-avibactam | 95.2 (180/189) | 94.2 (129/137) | 94.2 (65/69) | 92.9 (39/42) | 96.2 (25/26) | 97.4 (37/38) | 100 (14/14) | 0.845 | 0.686 | >0.999 |
| E. coli (n = 50) | 82.0 (41/50) | 78.4 (29/37) | 76.5 (13/17) | 82.4 (14/17) | 66.7 (2/3) | 87.5 (7/8) | 100 (5/5) | | | |
| ECC (n = 44) | 100 (44/44) | 100 (44/44) | 100 (31/31) | 100 (13/13) | NA | NA | NA | | | |
| K. pneumoniae (n = 79) | 100 (79/79) | 100 (40/40) | 100 (14/14) | 100 (5/5) | 100 (21/21) | 100 (30/30) | 100 (9/9) | | | |

[a]The inoculum effect was investigated on CPE isolates with a cefiderocol MIC ≤ 4 mg/L or aztreonam-avibactam MIC ≤ 4/4 mg/L at standard inoculum.
[b]ECC, *Enterobacter cloacae* complex; MBL, metallo-β-Lactamase; NA, not applicable.

(11) and mutation in *cirA* siderophore receptor or its regulators (24, 25). Several previous studies reported that NDM-5 was the NDM variant correlated with cefiderocol resistance (11, 22, 26). NDM-5 showed increased carbapenemase activity than NDM-1 due to the V88L substitution (27). In our study, NDM-1 but not NDM-5 was the main NDM variant of cefiderocol nonsusceptible NDM CPE, including a cluster of five *E. cloacae* complex strain ST316. In addition, the *E. coli* strain ST167, ST410, and ST617 identified in our cefiderocol nonsusceptible NDM producers were the widely distributed sequence types across countries (28). Among them, ST410 had been identified as the predominant strain of NDM-positive *E. coli* isolates in our institute (29). Although our investigation of the epidemiologic relationship among the cefiderocol nonsusceptible *E. coli* or *E. cloacae* complex isolates did not support a hospital outbreak, the increase in prevalence of cefiderocol nonsusceptibility among *Enterobacteriaceae* via the dissemination of certain bacterial clones is worthy of potential concern.

MBL-producing *E. cloacae* complex showed lower susceptibility to cefiderocol compared to MBL-producing *E. coli* or *K. pneumoniae* in our study. These cefiderocol nonsusceptible *E. cloacae* complex carried IMP-8 or NDM-1, and six different ST were identified among them. To date, no specific sequence type of *E. cloacae* complex was associated with cefiderocol resistance. *E. cloacae* complex could harbor a variety of antibiotic resistance gene, and epidemic carbapenem-resistant clones of *E. cloacae* complex such as ST78, ST171, and ST114 had emerged (30, 31). Therefore, the trend of cefiderocol nonsusceptibility in *E. cloacae* complex should be monitored further after the wide use of this novel agent.

ATM-AVI demonstrated potent *in vitro* activity against MBL CPE or cefiderocol nonsusceptible CPE. In one systemic review, 20.4% of MBL-producing *Enterobacterales* had MIC >4 mg/L for ATM when combined with AVI or CAZ-AVI, and 31 (93.9%) of them carried NDM carbapenemase (32). The susceptibility to ATM-AVI has not been compared between different MBLs. Our study showed that NDM-positive *E. coli* had the highest $MIC_{50}$ of ATM-AVI compared to VIM CPE, IMP CPE, or other species of NDM CPE. In addition, a cluster of four ATM-AVI nonsusceptible NDM-5-positive *E. coli* ST617 was found. NDM-5-positive *E. coli* with reduced susceptibility to ATM-AVI had been discovered worldwide, and molecular typing showed diverse clonal background (33). More than 10 sequence types were identified, including ST617 (33). The spread of these ATM-AVI-resistant clones could be a concern in clinical practice. Although ATM-AVI showed promising *in vitro* activity against MBL CPE and may serve as an option for cefiderocol nonsusceptible *Enterobacteriaceae*, cautions should be taken while prescribing ATM-AVI in area with high prevalence of NDM producers, especially when antimicrobial susceptibility test for ATM-AVI was not available.

The inoculum size of CPE substantially impacted the MICs of cefiderocol. In studies by Hobson et al. and Danjean et al., the inoculum effect of cefiderocol against CPE occurred in 37/37 of KPC-producing *Enterobacteriaceae* and 39/40 of CPE (including 11 NDM- and eight VIM-producing isolates), respectively (16, 17). Our results were consistent with theirs that 95.9% of CPE exhibited inoculum effect on cefiderocol. Moreover, we found inoculum effect on cefiderocol more commonly occurred among MBL CPE isolates than KPC CPE. For ATM-AVI, 95.2% susceptible CPE exhibited inoculum effect in our study, which was much higher than the report by Kim et al. (15/35, 42.9%) (18). The reason leading to this discrepancy was unclear. However, both of ours and Kim's studies revealed that carbapenemase-producing *K. pneumoniae* isolates exhibited a higher rate of ATM-AVI inoculum effect than *E. coli*. In the literature, variations in PBP3 enzymes and carbapenemase activity, or the production of other $\beta$-lactamases among different species of *Enterobacteriaceae* might all affect the magnitude of inoculum effect (14, 34, 35). The presence of inoculum effects suggested potential benefits of higher doses or combination therapy to combat CPE infections with high bacterial burden. *In vivo* or clinical studies are warrants to confirm the correlation between high inoculum and failure of cefiderocol or ATM-AVI monotherapy. In most cases, amikacin, tigecycline, or colistin would be the options of companion drug for CPE infections if based on *in vitro* susceptibilities. Further investigations are also needed to identify the superior combination regimen for MBL CPE infections.

This study has several limitations. First, the CPE isolates were from one single center, thus the distribution of bacteria species and MBL types may not be representative. Second, we did not investigate other molecular mechanisms which were associated with elevated cefiderocol or ATM-AVI MICs, or the *β*-lactamases that might impact the inoculum effect. Third, not all CPE isolates were tested for their MICs at high inoculum. The inoculum effect of CPE against cefiderocol or ATM-AVI might be more significant if all isolates were included.

In conclusion, cefiderocol and ATM-AVI showed potent *in vitro* activity against MBL-producing *Enterobacteriaceae*. NDM producer had lower susceptibility to cefiderocol and ATM-AVI compared to VIM or IMP producers. For most CPE isolates, inoculum effects on cefiderocol and ATM-AVI were observed. Our results suggested a potential risk of microbiological failure when using monotherapy with these newer antibiotics to treat CPE infections with high bacteria burden.

## MATERIALS AND METHODS

**Hospital setting and bacterial isolates.** This study was conducted at National Taiwan University Hospital (NTUH), a 2,200-bed tertiary care center in Northern Taiwan. At NTUH, nonduplicate meropenem-resistant (MIC $\geq$4 mg/L, according to Clinical and Laboratory Standards Institute [CLSI] 2021 criteria [36]) *Enterobacteriaceae* isolates were prospectively screened for the carriage of carbapenemase-encoding genes by PCR since 2013 based on NTUH infection control policy, and isolates positive for carbapenemase-encoding genes were stored. The susceptibility to meropenem was determined with Vitek 2 automated system (BioMérieux, Marcy l'Etoile, France).

In the present study, all *Enterobacteriaceae* carrying MBL genes or OXA-48-like genes, and randomly selected KPC-producing *Enterobacteriaceae* (with a one in 10 ratio) collected during 2016 to 2021 at NTUH were included and subjected to *in vitro* susceptibility studies. Isolates carrying dual-carbapenemase genes were excluded from the analysis. These *Enterobacteriaceae* isolates were isolated from various types of clinical specimens and anal screening cultures. We also recorded the specimen collection date, the type and location of patient ward, and the comorbidities of the source patient. The species identification were performed with Bruker Biotyper matrix assisted laser desorption ionization-time of flight mass spectrometry (MALDI-TOF MS) system (Bruker Corporation, Billerica, MA, USA). The study was approved by the NTUH Research Ethics Committee (registration number 201912161RINC) and written or oral informed consent was waived.

**Detection of carbapenemase-encoding genes.** Bacterial DNA was extracted using Viogene DNA extraction kit (Blood Genomic DNA Extraction Midiprep System; Viogene, Taipei, Taiwan) following the manufacturer's instruction. Genes encoding for carbapenemases, including class A ($bla_{KPC}$), class B ($bla_{IMP}$, $bla_{VIM}$, $bla_{NDM}$), and class D ($bla_{OXA-48-like}$) were detected by PCR using the primers described previously (37, 38). All PCR amplicons were sequenced by Sanger sequencing. The sequencing results were compared to the National Center for Biotechnology Information (NCBI) database at https://blast.ncbi.nlm.nih.gov/Blast.cgi to determine the subtypes of carbapenemase. In this study, *Enterobacteriaceae* isolates positive for the carbapenemase-encoding genes by PCR tests were considered CPE.

**Molecular typing.** Pulsed-field gel electrophoresis (PFGE) was performed as previously described method (29). Genomic DNA of bacterial isolates were prepared in agarose plugs and digested with the restriction enzyme Spel (New England Biolabs, NEB, Beverly, MA, USA). The PFGE banding patterns were analyzed by BioNumerics software version 8.0 (Applied Maths, Sint-Martens-Latem, Belgium). Isolates that exhibited similarity of $\geq$80% of their banding patterns were considered as closely related strains. Multilocus sequence typing (MLST) with housekeeping genes of *E. coli* (including adk, fumC, gyrB, icd, mdh, purA, and recA) and *Enterobacter cloacae* (including dnaA, fusA, gyrB, leuS, pyrG, rplB, and rpoB) were performed according to previously described protocol (39–41). Information from the PubMLST website https://pubmlst.org/organisms were used to determine the sequence type (ST) of *E. coli* and *E. cloacae* isolates.

**Antimicrobial susceptibility testing.** The MICs of each CPE isolate to amikacin, aztreonam, ciprofloxacin, colistin, and tigecycline were determined by commercialized broth microdilution using Sensititre 96-well plates (GNX2F; Trek Diagnostic Systems, East Grinstead, United Kingdom). The MICs of each CPE isolate to cefiderocol and ATM-AVI were determined by broth microdilution method and the results were interpreted according to Clinical and Laboratory Standards Institute (CLSI) 2021 criteria (36). Cefiderocol was tested using iron-depleted cation-adjusted Mueller-Hinton broth prepared following the CLSI guidance (36). Cefiderocol and avibactam were obtained from Shionogi & Co., Ltd. (Osaka, Japan). Currently, ATM-AVI does not have approved susceptibility breakpoints for *Enterobacteriaceae*, thus the CLSI susceptible breakpoint for aztreonam (MIC $\leq$ 4 mg/L) for *Enterobacteriaceae* was taken as reference to interpret the results at with a fixed concentration of avibactam at 4 mg/L in this study. For tigecycline, the U.S. Food and Drug Administration susceptibility breakpoint (MIC $\leq$ 2 mg/L) for *Enterobacteriaceae* was used. For isolates that were susceptible to cefiderocol or having an ATM-AVI MIC $\leq$ 4/4 mg/L at standard inoculum ($10^5$ CFU [CFU]/mL), the MICs at high inoculum ($10^7$ CFU/mL) would be determined. The *in vitro* inoculum effect was defined as a $\geq$8-fold MIC increase at the high inoculum (14).

**Statistical analysis.** The difference in antimicrobial susceptibility or inoculum effect between groups were compared using the chi-squared test or Fisher's exact test. All *P* values were two-sided, and a *P*

value <0.05 was considered statistically significant. Statistical analyses were performed using Stata software (version 11; StataCorp, College Station, TX, USA).

**Ethical approval statement.** The study was approved by the NTUH Research Ethics Committee (registration number 201912161RINC) and written or oral informed consent was waived.

**Data availability.** Deidentified participant-level data will be available on publication of the study. Requests for data should be sent to the corresponding author by email and, on review of the proposed protocol and signing of a data sharing agreement, the data will be made available.

## SUPPLEMENTAL MATERIAL

Supplemental material is available online only.

**SUPPLEMENTAL FILE 1**, DOCX file, 0.6 MB.

## ACKNOWLEDGMENTS

The study was supported by the National Taiwan University Hospital, Taipei, Taiwan (grant number NTUH110-35) and Ministry of Science and Technology, Taiwan (grant number MOST110-2314-B-002-251). The sponsors had no role in the study design, data collection and analysis, manuscript preparation, or the decision to submit for publication.

The authors have no conflicts of interest to declare.

Conceived and designed the experiments: Y.-S.H.; acquisition of data: Y.-S.H., P.-C.C.; analysis of data: Y.-S.H., P.-C.C., P.-Y. C.; drafting and revision of the manuscript: Y.-S.H., J.-T. W.

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
