## [Reviewer comments · Microbiology Spectrum]

Microbiology Spectrum

In Vitro Activities and Inoculum Effects of Cefiderocol and Aztreonam-avibactam against Metallo- β -lactamase-producing *Enterobacteriaceae*

Yu-Shan Huang, Pao-Yu Chen, Pei-Chun Chou, and Jann-Tay Wang

Corresponding Author(s): Jann-Tay Wang, National Taiwan University Hospital

Review Timeline:

Submission Date:	February 10, 2023
Editorial Decision:	March 26, 2023
Revision Received:	April 14, 2023
Accepted:	April 18, 2023

Editor: Pablo Power

Reviewer(s): Disclosure of reviewer identity is with reference to reviewer comments included in decision letter(s). The following individuals involved in review of your submission have agreed to reveal their identity: Dennis Nurjadi (Reviewer #2)

Transaction Report:

DOI: <https://doi.org/10.1128/spectrum.00569-23>

March 26, 2023

Prof. Jann-Tay Wang
National Taiwan University Hospital
Department of Internal Medicine
No. 7 Chung-Shan S rd, Taipei, Taiwan
Taipei, -- Not Applicable --
Taiwan

Re: Spectrum00569-23 (In Vitro Activities and Inoculum Effects of Cefiderocol and Aztreonam-avibactam against Metallo- β -lactamase-producing *Enterobacteriaceae*)

Dear Prof. Jann-Tay Wang:

Link Not Available

Sincerely,

Pablo Power

Journals Department
Reviewer comments:

Reviewer #1 (Comments for the Author):

This is a nice study demonstrating what we already suspect to be the case. ATM AVI is predicted to be more potent than FDC and authors show this. Molecular data would make the paper stronger.

The paper could be shortened without losing much of the message. There was some redundancy

Reviewer #2 (Comments for the Author):

The authors investigated the association between the inoculum effect on cefiderocol and aztreonam-avibactam MIC using isolates collected through microbiological diagnostics focusing on the various types of beta-lactamases/MBLs. The study's premise is interesting since this contributes to knowledge on newer antibiotics, which is not yet widely used. The text-to-information/results ratio can be improved, especially the discussion can be shortened for a more concise and crisp discussion. The quality of English needs attention; some sentences appeared disjointed, disrupting the overall text flow. Some information was presented abruptly without apparent reference to the preceding sentences. For example, the beta-lactamase subtypes were not mentioned in the results section and tables. But, in Figure 3 and the discussion, NDM-1, NDM-5 and KPC-2 were specified. Most PCR-based detection methods cannot differentiate the subtypes due to the minor mutations/deviations between the subtypes.

Specific comments:

- Which breakpoints were used to interpret the susceptibility testing for meropenem? Also CLSI?
- Isolates included in the study: were these clinical isolates? Or screening/colonization? This was not clearly stated.
- Methods: you did not specify that the detection method can differentiate the subtypes of beta-lactamases, yet you specified NDM-1 and NDM-5 in the text; this requires some explanation.
- Line 354: Please correct to "chi-squared test". "chai" is a type of tea-based drink and not a statistical method.
- Line 72-74: The most prevalent carbapenemase in many European countries is OXA-48. KPC is often detected, but you cannot generalize that KPC is the dominant mechanism of carbapenem resistance in Europe. However, there has been a sharp increase in NDM-1 since 2022, notably due to migration and the war in the Ukraine.
- Line 89-92: please check this sentence (grammar); in this case, please do not start the sentence with "As" - this is unnecessary. Consider adding the citations of the studies showing the resistance to the mentioned antibiotics. It is sufficient to cite the first studies showing in vivo resistance development to these agents.
- Paragraph starting with line 93: consider beginning the sentence with "The MIC of many beta-lactam antibiotics can be influenced/affected by the inoculum effect..." and then begin the explanation. This may be the better option for the better fluidity of reading (text flow).
- Lines 109-110: do you have any information on the subtypes of the metallo-beta-lactamases? NMD-1, NMD-5, VIM-1, VIM-2 etc? I assume the PCR panel used to detect these genes, cannot differentiate between the subtypes? But in line 150, you mentioned the subtypes of the NDM of *E. coli* (NDM-1 and NDM-5). If you have this information, please include it in the results. This would have been interesting since NDM-5 has been shown to have stronger hydrolyzing effects on cefiderocol than NDM-1, despite being very low.
- Lines 120-121: please add the absolute numbers (n/N) to the percentages.. this applies throughout the results section. There are some results which were displayed as n/N, % and some were only percentages, and the information displayed should be consistent throughout.
- Lines 194-195: There are also reports on the presence of extrachromosomal fec operon, which can, in conjunction with "weaker" MBL, such as VIM, result in a resistance phenotype (<https://pubmed.ncbi.nlm.nih.gov/36245258/>). You may want to include this information.
- Line 202: Unfortunately, I was not able to access this publication, but did the authors of the paper check for cirA or any siderophore mutations? Since this assumption is otherwise bold. In the abstract, I cannot derive if they actually demonstrated that the cefiderocol was mediated by OXA-181 - if not, please, this sentence should be removed.
- Lines 210-211: Please cite the original paper showing this. NDM-5 differ in positions 88 (Val→Leu) and 154 (Met→Leu) from NDM-1. I am not aware that the V88L mutation is the cause of stronger cefiderocol hydrolysis. If you have the publication showing this, then please correct the citation. Otherwise it is always recommendable to cite the original paper than reviews, since the data of the review may be incomplete.
- 216-221: could it be that these isolates originated from a hospital outbreak/transmissions? Was there any epidemiological overlap between these patients? Some data on the origin of the isolates/samples should be included in the methods.
- Lines 236-238: these are not coherent. In the first sentence, you mentioned MBLs, but in the second sentence CPE.. maybe you should add..."NDM... higher than x, y, z type MBL" for better readability.

- Line 237-242: The discussion is partly confusing. Since random studies are mentioned without a clear reference/flow of information, for example, reference 32 appeared out of nowhere and did not relate directly to the preceding sentences.
- Figure 2 is redundant to the Tables and can be moved to supplementary data.
- Figure 3: it would be nice if some statements could be made about the potential reasons for clonality, i.e. transmission/outbreak? Otherwise, this may exaggerate the interpretation of the results presented.

Reviewer #3 (Comments for the Author):

In this manuscript, the author evaluated in vitro activities of Cefiderocol and Aztreonam-avibactam against Metallo- β -lactamase-producing Enterobacteriaceae, and further investigated the impact of inoculum size. Overall, this manuscript is well organized and well written. It's interesting that the inoculum size of CPE substantially impacted the MICs of cefiderocol and aztreonam-avibactam, which can be addressed in the future research.

Staff Comments:

Preparing Revision Guidelines

Please return the manuscript within 60 days; if you cannot complete the modification within this time period, please contact me. If you do not wish to modify the manuscript and prefer to submit it to another journal, please notify me of your decision immediately so that the manuscript may be formally withdrawn from consideration by Microbiology Spectrum.

Apr 11th, 2023

Manuscript Number: Spectrum 00569-23R1

Title: **In Vitro Activities and Inoculum Effects of Cefiderocol and Aztreonam-avibactam against Metallo- β -lactamase-producing *Enterobacteriaceae***

Dear Editor,

Thank you for your kind consideration of our manuscript. We have carefully read and responded point by point to the questions and comments of the reviewers. The response to each reviewer's comment has been incorporated into the revised manuscript and is underlined. Figure 2 has been changed to Supplementary Figure 1, and Supplementary Table 1 has been added as suggested by the reviewers. We appreciate their thoughtful and helpful review, which helped to strengthen the manuscript. Attached are point-by-point responses to each of the comments and the revised manuscript. We hope that the manuscript is now acceptable for publication. Please do not hesitate to contact me if there are any questions.

Sincerely,

Jann-Tay Wang, M.D., Ph.D.

Department of Internal Medicine, National Taiwan University Hospital, No.7, Chung Shan S. Rd., Zhongzheng Dist., Taipei City 10002, Taiwan.

E-mail address: wangjt1124@ntu.edu.tw

Tel: +886-2-23123456 ext. 65054; Fax:+886-2-23971412;

Responses to the Reviewers

Reviewers' Comments

Comments of reviewer #1:

Reviewer #1: This is an nice study demonstrating what we already suspect to be the case. ATM AVI is predicted to be more potent than FDC and authors show this. Molecular data would make the paper stronger. The paper could be shortened without loosing much of the message. There was some redundancy.

Reply: Thank you for your comment. We have shortened the Results and Discussion sections to avoid redundancy. The deleted sentences are listed below. Also, the Figure 2 was changed to Supplementary Figure 1 as reviewer 2's suggestion.

The deleted sentences:

- Lines 134-138: "Among MBL CPE, NDM producers showed lower susceptibility and higher MIC₅₀/MIC₉₀ values of cefiderocol (78.4% [58/74], 2/16 mg/L) than IMP (92.9% [39/42], 0.375/4 mg/L) and VIM (96.3% [26/27], 1/4 mg/L) producers. ~~The MIC₅₀/MIC₉₀s of cefiderocol for KPC CPE and OXA-48 like CPE were both 0.75/4 mg/L."~~
- Lines 143-146: "Supplementary Figure 1A demonstrates cefiderocol MIC distributions for different species of MBL CPE. ~~NDM positive *E. coli*, NDM positive *E. cloacae complex*, and IMP positive *E. cloacae complex* displayed high MIC₉₀s of cefiderocol (16 mg/L), and their susceptibility rates to cefiderocol were 72.7% (16/22), 77.4% (24/31), and 76.9% (10/13), respectively (Table 1). Overall, MBL-producing *E. cloacae complex* were less susceptible to cefiderocol than MBL-producing *E. coli* or *K. pneumoniae* isolates (77.3% [34/44] vs. 86.0% [37/43] and 95.0% [38/40])."~~
- Lines 213-216: "These results indicated that the presence of transferable

resistance determinant or other β -lactamases among MBL CPE might affect their susceptibilities to ATM-AVI or cefiderocol. Noteworthy, 8% of KPC and OXA-48-like CPE were non-susceptible to cefiderocol in our study. *Enterobacteriaceae* isolates with mutant KPC had been shown to display higher cefiderocol MIC compared to those with the ancestral allele. Although OXA-48-like carbapenemase was not correlated to cefiderocol resistance, its derivative OXA-181 had been found to be carried by a cefiderocol-resistant *K. pneumoniae* which also carried KPC-125”

- Lines 217-221: “Isolates harboring NDM carbapenemases demonstrated higher cefiderocol MICs than other CPE. This finding was in line with previous reports (22, 23). ~~In a review obtaining MIC data for Enterobacterales from 15 studies, the pooled susceptibility rate of NDM CPE to cefiderocol was 83.4%. Mechanism responsible for cefiderocol resistance among NDM CPE included increased expression of blaNDM gene (11) and mutation in cirA siderophore receptor or its regulators (24, 25).”~~
- Lines 222-223: “NDM-5 showed increased carbapenemase activity than NDM-1 due to the V88L substitution (27), ~~and the presence of NDM-5 in cirA mutants of *E. cloacae* enhanced cefiderocol resistance”~~
- Lines 236-238: “These cefiderocol non-susceptible *E. cloacae complex* carried IMP-8 or NDM-1, and six different ST were identified among them. ~~The mechanisms for cefiderocol resistance in *E. cloacae complex* included mutation in the R2 loop of AmpC beta-lactamase or in siderophore receptors (23, 25, 29).~~ To date, no specific sequence type of *E. cloacae complex* was associated with cefiderocol resistance.”

Comments of reviewer #2:

Reviewer #2: The authors investigated the association between the inoculum effect on cefiderocol and aztreonam-avibactam MIC using isolates collected through microbiological diagnostics focusing on the various types of beta-lactamases/MBLs. The study's premise is interesting since this contributes to knowledge on newer antibiotics, which is not yet widely used. The text-to-information/results ratio can be improved, especially the discussion can be shortened for a more concise and crisp discussion. The quality of English needs attention; some sentences appeared disjointed, disrupting the overall text flow. Some information was presented abruptly without apparent reference to the preceding sentences. For example, the beta-lactamase subtypes were not mentioned in the results section and tables. But, in Figure 3 and the discussion, NDM-1, NDM-5 and KPC-2 were specified. Most PCR-based detection methods cannot differentiate the subtypes due to the minor mutations/deviations between the subtypes.

Reply: Thank you for your comment. We have shortened the Results and Discussion sections, as listed below. Also, we rewrite the paragraph to improve readability. The detection method to differentiate carbapenemase subtypes was added, as shown in the response to comment 3.

The deleted sentences:

- Lines 134-138: "Among MBL CPE, NDM producers showed lower susceptibility and higher MIC₅₀/MIC₉₀ values of cefiderocol (78.4% [58/74], 2/16 mg/L) than IMP (92.9% [39/42], 0.375/4 mg/L) and VIM (96.3% [26/27], 1/4 mg/L) producers. ~~The MIC₅₀/MIC₉₀s of cefiderocol for KPC CPE and OXA-48-like CPE were both 0.75/4 mg/L."~~
- Lines 143-146: "Supplementary Figure 1A demonstrates cefiderocol MIC

distributions for different species of MBL CPE. ~~NDM-positive *E. coli*, NDM-positive *E. cloacae complex*, and IMP-positive *E. cloacae complex*~~ displayed high MIC_{90s} of cefiderocol (16 mg/L), and their susceptibility rates to cefiderocol were 72.7% (16/22), 77.4% (24/31), and 76.9% (10/13), respectively (Table 1). Overall, MBL-producing *E. cloacae complex* were less susceptible to cefiderocol than MBL-producing *E. coli* or *K. pneumoniae* isolates (77.3% [34/44] vs. 86.0% [37/43] and 95.0% [38/40]).”

- Lines 213-216: “These results indicated that the presence of transferable resistance determinant or other β-lactamases among MBL CPE might affect their susceptibilities to ATM-AVI or cefiderocol. ~~Noteworthy, 8% of KPC and OXA-48 like CPE were non-susceptible to cefiderocol in our study. Enterobacteriaceae isolates with mutant KPC had been shown to display higher cefiderocol MIC compared to those with the ancestral allele. Although OXA-48 like carbapenemase was not correlated to cefiderocol resistance, its derivative OXA-181 had been found to be carried by a cefiderocol-resistant *K. pneumoniae* which also carried KPC-125”~~
- Lines 217-221: “Isolates harboring NDM carbapenemases demonstrated higher cefiderocol MICs than other CPE. This finding was in line with previous reports (22, 23). ~~In a review obtaining MIC data for Enterobacterales from 15 studies, the pooled susceptibility rate of NDM CPE to cefiderocol was 83.4%. Mechanism responsible for cefiderocol resistance among NDM CPE included increased expression of blaNDM gene (11) and mutation in cirA siderophore receptor or its regulators (24, 25).”~~
- Lines 222-223: “NDM-5 showed increased carbapenemase activity than NDM-1 due to the V88L substitution (27), ~~and the presence of NDM-5 in cirA mutants of *E. cloacae* enhanced cefiderocol resistance”~~

- Lines 236-238: “These cefiderocol non-susceptible *E. cloacae* complex carried IMP-8 or NDM-1, and six different ST were identified among them. ~~The mechanisms for cefiderocol resistance in *E. cloacae* complex included mutation in the R2 loop of AmpC beta-lactamase or in siderophore receptors (23, 25, 29).~~ To date, no specific sequence type of *E. cloacae* complex was associated with cefiderocol resistance.”

Specific comments

Comment 1. Which breakpoints were used to interpret the susceptibility testing for meropenem? Also CLSI?

Reply: The 2021 CLSI breakpoint was used to interpret the susceptibility testing for meropenem. We added this information in the method (line 299-300, underlined): “At NTUH, non-duplicate meropenem-resistant (MIC \geq 4 mg/L, according to Clinical and Laboratory Standards Institute [CLSI] 2021 criteria) *Enterobacteriaceae* isolates were prospectively screened for the carriage of carbapenemase-encoding genes.”

Comment 2. Isolates included in the study: were these clinical isolates? Or screening/colonization? This was not clearly stated.

Reply: Thank you for your query. The *Enterobacteriaceae* isolates in this study include both isolates from clinical specimen and anal screening culture. We revised the description in the method to clarify this point (line 310-311, underlined): “These *Enterobacteriaceae* isolates were isolated from various types of clinical specimens and anal screening cultures.”

We also revised the result section to provide further details about the source of *Enterobacteriaceae* isolates (line 124-126, underlined): “The majority of CPE were clinical isolates from urine (n=68, 34.9%), blood (n=50, 25.6%), sputum (n=37, 19.0%),

skin pus or wound (n=14, 7.2%), body fluid (n=5, 2.6%), and catheter tip (n=1, 0.5%).
Twenty (10.3%) of the 195 CPE were from anal screening cultures.”

Comment 3. Methods: you did not specify that the detection method can differentiate the subtypes of beta-lactamases, yet you specified NDM-1 and NDM-5 in the text; this requires some explanation.

Reply: We apologize for the lack of the information on the differentiation of carbapenemase subtypes. We added one sentence to describe the detection method (lines 324-327, underlined): “All PCR amplicons were sequenced by Sanger sequencing. The sequencing results were compared to the National Center for Biotechnology Information (NCBI) database at <https://blast.ncbi.nlm.nih.gov/Blast.cgi> to determine the subtypes of carbapenemase.”

Comment 4. Line 354: Please correct to "chi-squared test". "chai" is a type of tea-based drink and not a statistical method.

Reply: Thank you for pointing out the error. We corrected "chai-squared test" to "chi-squared test" (line 367).

Comment 5. Line 72-74: The most prevalent carbapenemase in many European countries is OXA-48. KPC is often detected, but you cannot generalize that KPC is the dominant mechanism of carbapenem resistance in Europe. However, there has been a sharp increase in NDM-1 since 2022, notably due to migration and the war in the Ukraine.

Reply: Thank you for your comment. We agree with the reviewer that the description of carbapenemase types in different regions should be more

comprehensive. The sentences were revised as follow (line 71-75, underlined): “In Southeast Asia, metallo-β-lactamases (MBLs) and oxacillinase-48 (OXA-48) were the main carbapenemases in *Enterobacteriaceae*, whereas *Klebsiella pneumoniae* carbapenemase (KPC) was the most common carbapenemase in North America (2). In Europe, OXA-48 and KPC were more frequently detected than MBLs, but an increase in NDM-1 producers was noted since 2022, which was associated with the migration of people (3,4).” Two reference papers (ref 3 and 4) were cited here:

- **Grundmann H, et al. Lancet Infect Dis. 2017;17(2):153-163:** This study reported data on the occurrence of carbapenemase-producing *K. pneumoniae* and *E. coli* across Europe.
- **Sandfort M, et al. Euro Surveill. 2022;27(50):2200926:** This study reported the rapidly increasing numbers of NDM-1- and NDM-1/OXA-48-producing *K. pneumoniae* in Germany, and, among these cases, a presence in Ukraine before diagnosis was frequently reported.

Comment 6. Line 89-92: please check this sentence (grammar); in this case, please do not start the sentence with "As" - this is unnecessary. Consider adding the citations of the studies showing the resistance to the mentioned antibiotics. It is sufficient to cite the first studies showing in vivo resistance development to these agents.

Reply: Thank you for your suggestions. We revised the sentence and cited the first studies reporting the resistance of CPE to cefideocol or ATM-AVI (line 90-92, underlined): “Data on the susceptibilities of multidrug-resistant *Enterobacteriaceae* to cefideocol or ATM-AVI have been increasingly reported (11, 12). However, many did not compare the results across bacterial species or types of MBL.”

We cited two reference papers which were the first or early reports of cefideocol

or ATM-AVI resistance in *Enterobacteriaceae* (Simner PJ, et al. *Clin Infect Dis.* 2022;75(1):47-54, Alm RA, et al. *Antimicrob Chemother.* 2015;70(5):1420-8).

Comment 7. Paragraph starting with line 93: consider beginning the sentence with "The MIC of many beta-lactam antibiotics can be influenced/affected by the inoculum effect..." and then begin the explanation. This may be the better option for the better fluidity of reading (text flow).

Reply: Thank you for your suggestion. We revised the sentence as follows (line 94-96, underlined): "The MIC of many beta-lactam antibiotics can be influenced by the inoculum effect. Inoculum effect refers to a laboratory phenomenon in which the MIC of an antibiotic increases significantly when the number of inoculated organisms is increased."

Comment 8. Lines 109-110: do you have any information on the subtypes of the metallo-beta-lactamases? NMD-1, NMD-5, VIM-1, VIM-2 etc? I assume the PCR panel used to detect these genes, cannot differentiate between the subtypes? But in line 150, you mentioned the subtypes of the NDM of *E. coli* (NDM-1 and NDM-5). If you have this information, please include it in the results. This would have been interesting since NDM-5 has been shown to have stronger hydrolyzing effects on cefiderocol than NDM-1, despite being very low.

Reply: Thank you for your query. We had sequenced the PCR amplicons of carbapenemases to determine their subtype. Data on MBL, KPC, and OXA-48-like subtypes are available. We added a paragraph in the result section to present this information (line 114-119, underlined): "Of the 74 NDM-producers, 49 (66.2%) were NDM-1, 24 (32.4%) were NDM-5, and one (1.4%) was NDM-4. All IMP-producers were IMP-8, and all VIM-producers were VIM-1. The subtypes of KPC carbapenemase

included KPC-2 (89.5%, 34/38), KPC-17 (5.3%, 2/38), KPC-3 (2.6%, 1/38), and KPC-65 (2.6%, 1/38). Of the 14 OXA-48-like CPE, 13 were OXA-48 and one was OXA-181.”

Comment 9. Lines 120-121: please add the absolute numbers (n/N) to the percentages. this applies throughout the results section. There are some results which were displayed as n/N, % and some were only percentages, and the information displayed should be consistent throughout.

Reply: Thank you for your comment. We added absolute numbers (n/N) to the percentages throughout the results section. The revised sentences are listed, as follows (underlined):

- Lines 131-137 (underlined): “The susceptibility rate of all CPE to amikacin, aztreonam, ciprofloxacin, and tigecycline were 92.3% (180/195), 21.5% (42/195), 24.1% (47/195), and 99.0% (193/195) respectively. 87.7% (171/195) of CPE isolates had a colistin MIC \leq 2 mg/L. The susceptible rates of MBL, KPC and OXA-48-like CPE to cefiderocol were 86.0% (123/143), 92.1% (35/38) and 92.9% (13/14), respectively. Among MBL CPE, NDM producers showed lower susceptibility and higher MIC₅₀/MIC₉₀ values of cefiderocol (78.4% [58/74], 2/16 mg/L) than IMP (92.9% [39/42], 0.375/4 mg/L) and VIM (96.3% [26/27], 1/4 mg/L) producers.
- Lines 144-146 (underlined): “Overall, MBL-producing *E. cloacae* complex were less susceptible to cefiderocol than MBL-producing *E. coli* or *K. pneumoniae* isolates (77.3% [34/44] vs. 86.0% [37/43] and 95.0% [38/40]).”
- Lines 147 (underlined): “ATM-AVI inhibited 95.8% (137/143) of MBL CPE and 100% of KPC CPE and OXA-48-like CPE at the 4/4 mg/L breakpoint.”
- Lines 153 (underlined): “NDM- and VIM-positive *E. coli* showed elevated MIC_{90s} (8 mg/L and 16 mg/L, respectively) and lower susceptibility to ATM-AVI (77.3%

[17/22] and 75.0% [3/4], respectively) when compared to MBL CPE of other species.

- Lines 184-187 (underlined): “The inoculum effect of cefiderocol and ATM-AVI were observed in 98.4% (121/123) and 94.2% (129/137) of MBL CPE isolates, respectively. Compared to MBL CPE, KPC CPE showed a significantly lower rate of inoculum effect for cefiderocol (98.4% [121/123] vs. 88.6% [31/35], p=0.022).”
- Lines 191-193 (underlined): “inoculum effect was less commonly detected in carbapenemase-producing *E. coli* (82.0%, 41/50) than in carbapenemase-producing *E. cloacae* complex (100%, 44/44) or *K. pneumoniae* (100%, 79/79) (Table 2).

Comment 10. Lines 194-195: There are also reports on the presence of extrachromosomal fec operon, which can, in conjunction with "weaker" MBL, such as VIM, result in a resistance phenotype (<https://pubmed.ncbi.nlm.nih.gov/36245258/>). You may want to include this information.

Reply: Thank you for your suggestion. We added this information in the discussion section (Lines 211-214, underlined): “The detailed β -lactamases profile of MBL-producers in Terrier’s work was not provided, but in another study published by the same group, presence of extended-spectrum β -lactamases such as BEL and SHV conferred a greater impact on cefiderocol MICs than ATM-AVI MICs among *E. coli* (20). Furthermore, Kocer et al. showed that the acquisition of extrachromosomal fec operon in conjunction with VIM-type MBL could lead to a significant elevation of the cefiderocol MIC (21). These results indicated that the presence of transferable resistance determinant or other β -lactamases among MBL CPE might affect their susceptibilities to ATM-AVI or cefiderocol.” The suggested reference paper was cited.

Comment 11. Line 202: Unfortunately, I was not able to access this publication, but did the authors of the paper check for cirA or any siderophore mutations? Since this assumption is otherwise bold. In the abstract, I cannot derive if they actually demonstrated that the cefiderocol was mediated by OXA-181 - if not, please, this sentence should be removed.

Reply: In the study by Gaibani P et al., they performed whole genome sequence of *Klebsiella pneumoniae* BO714 coproducing KPC and OXA-181 carbapenemase. The author reported that BO714 chromosome harbored virulence factors such as iron acquisition systems (iutA), but they did not check for cirA or siderophore mutations. Therefore, we removed this sentence and the related reference as suggested by the reviewer.

Comment 12. Lines 210-211: Please cite the original paper showing this. NDM-5 differ in positions 88 (Val→Leu) and 154 (Met→Leu) from NDM-1. I am not aware that the V88L mutation is the cause of stronger cefiderocol hydrolysis. If you have the publication showing this, then please correct the citation. Otherwise it is always recommendable to cite the original paper than reviews, since the data of the review may be incomplete.

Reply: Thank you for your suggestion. We cited the paper which discovered NDM-5 and reported its difference from NDM-1 (**Hornsey M et al., Antimicrob Agents Chemother. 2011 Dec;55(12):5952-4**). We had searched the literature regarding cefiderocol resistance and did not find study showing the correlation between V88L mutation and enhanced cefiderocol hydrolysis.

Comment 13. 216-221: could it be that these isolates originated from a hospital

outbreak/transmissions? Was there any epidemiological overlap between these patients? Some data on the origin of the isolates/samples should be included in the methods.

Reply: Thank you for your comment. We included the epidemiologic data of the source patient to investigate the correlation of cefiderocol or ATM-AVI non-susceptible CPE. The method was revised (line 311-312, underlined): “These *Enterobacteriaceae* isolates were isolated from various types of clinical specimens and anal screening cultures. We also recorded the specimen collection date, the type and location of patient ward, and the comorbidities of the source patient.”

As suggested by the reviewer, we analyzed the epidemiologic relationship of the 21 cefiderocol or ATM-AVI non-susceptible *E. coli* and *E. cloacae complex* isolates, which were the isolates subjected to PFGE analysis (data shown in the original Figure 3). The data are presented in the **Supplementary Table 1** (as shown below). Although some epidemiologic correlation was found between two ATM-AVI non-susceptible ST617 *E. coli* isolates and among three cefiderocol non-susceptible ST316 *E. cloacae complex*, it was not sufficient to support a hospital outbreak. We added several sentences in the result section to describe the findings:

- Line 168-171, underlined: “Two of the four ST617 *E. coli* isolates (M119 and M137) were from patients who had recently been discharged from two adjacent hospital wards within a two-month period (Supplementary Table 1).”
- Line 174-177, underlined: “Analysis of the epidemiologic data showed that three ST316 *E. cloacae complex* isolates (M89, M101, M124) were from patients who had been admitted to the hematology ward within the same year, but their admission periods did not overlap (Supplementary Table 1).”

We also revised the discussion section (line 223-231, underlined) to address this finding: “In our study, NDM-1 but not NDM-5 was the main NDM variant of

cefiderocol non-susceptible NDM CPE, including a cluster of five *E. cloacae* complex strain ST316. In addition, the *E. coli* strain ST167, ST410, and ST617 identified in our cefiderocol non-susceptible NDM-producers were the widely distributed sequence types across countries. Among them, ST410 had been identified as the predominant strain of NDM-positive *E. coli* isolates in our institute. Although our investigation of the epidemiologic relationship among the cefiderocol non-susceptible *E. coli* or *E. cloacae* complex isolates did not support a hospital outbreak, the increase in prevalence of cefiderocol non-susceptibility among *Enterobacteriaceae* via the dissemination of certain bacterial clones is worthy of potential concern.”

Supplementary Table 1. Microbiological and epidemiological characteristics of the 21 cefiderocol or ATM-AVI non-susceptible *E. coli* and *E. cloacae* complex isolates

Isolate No	Species	ATM-AVI		CFD		Carbapenemase	ST	Date of Isolation	Specimen Type	Ward of CPE isolation
		MIC	SIR	MIC	SIR					
K031	E. coli	0.125	S	8	I	KPC-2	131	2020/8/3	Urine	ICU A
M052	E. coli	1	S	16	R	NDM-5	167	2018/5/13	Blood	General ward A
M104	E. coli	8	I	8	I	NDM-1	410	2020/8/14	Urine	General ward B
M011	E. coli	4	S	8	I	NDM-1	410	2016/07/30	Urine	General ward C
M010	E. coli	4	S	16	R	NDM-1	410	2016/6/21	Anal swab	General ward D
M055	E. coli	8	I	8	I	NDM-5	617	2018/7/14	Urine	ER
M136	E. coli	1	S	16	R	NDM-5	617	2020/12/15	Urine	General ward E
M119*	E. coli	8	I	4	S	NDM-5	617	2020/10/22	Urine	ER*
M137*	E. coli	8	I	2	S	NDM-5	617	2020/12/17	Urine	ER*
M078	E. coli	16	R	4	S	NDM-5	617	2019/11/12	Urine	ER
M004	E. coli	16	R	0.125	S	VIM-1	5229	2016/2/15	Anal swab	ICU B
M110	E. cloacae complex	0.25	S	16	R	IMP-8	25	2020/9/21	Sputum	General ward F
M113	E. cloacae complex	≤0.06	S	8	I	IMP-8	66	2020/10/9	Sputum	General ward G
M053	E. cloacae complex	0.5	S	16	R	NDM-1	90	2018/5/14	Sputum	General ward H
M103	E. cloacae complex	0.5	S	32	R	IMP-9	90	2020/8/8	Skin pus	General ward I
M089**	E. cloacae complex	0.25	S	8	I	NDM-1	316	2020/3/15	Anal swab	ICU C**
M155	E. cloacae complex	0.125	S	8	I	NDM-1	316	2021/7/1	Urine	General ward J
M085	E. cloacae complex	0.125	S	>32	R	NDM-1	316	2020/2/10	Urine	ICU D
M101**	E. cloacae complex	0.25	S	32	R	NDM-1	316	2020/7/25	Urine	General ward A**
M124**	E. cloacae complex	0.5	S	16	R	NDM-1	316	2020/11/9	Blood	General ward G**
M125	E. cloacae complex	≤0.06	S	8	I	NDM-1	1385	2020/11/11	Sputum	General ward K

Abbreviations: ATM-AVI, aztreonam-avibactam; CFD, cefiderocol; CPE, carbapenemase-producing *Enterobacteriaceae*; ER, emergency department; ICU, intensive care unit; MIC, minimum inhibitory concentration; SIR, susceptible, intermediate, or resistant; ST, sequence type

* Before presenting to the ER, the patient carrying M119 was admitted to ward L from 2020/07/07 to 2020/10/12, and the patient carrying M137 was admitted to ward M ward from 2020/11/22 to 2020/12/10. Wards L and M were two adjacent general wards on the same floor.

** The patient carrying M089 was admitted to ward A from 2020/01/18 to 2020/02/20 before being transferred to ICU, the patient carrying M101 was admitted to ward A from 2020/06/16 to 2020/07/28, and the patient carrying M124 was admitted to ward G from 2020/10/18 to 2020/11/23. Wards A and G were two adjacent wards on the same floor. They are the wards for patients with hematologic malignancies.

Comment 14. Lines 236-238: these are not coherent. In the first sentence, you mentioned MBLs, but in the second sentence CPE. maybe you should add..."NDM... higher than x, y, z type MBL" for better readability.

Reply: We rewrote this paragraph for better understanding (line 247-250, underlined): "In one systemic review, 20.4% of MBL-producing *Enterobacteriales* had MIC >4 mg/L for ATM when combined with AVI or CAZ-AVI, and 31 (93.9%) of them carried NDM carbapenemase (31). The susceptibility to ATM-AVI has not been compared between different MBLs. Our study showed that NDM-positive *E. coli* had the highest MIC₅₀ of ATM-AVI when compared to VIM CPE, IMP CPE, or other species of NDM CPE."

Comment 15. Line 237-242: The discussion is partly confusing. Since random studies are mentioned without a clear reference/flow of information, for example, reference 32 appeared out of nowhere and did not relate directly to the preceding sentences.

Reply: We rewrote this paragraph to improve the text flow. Following the sentences in line 247-250 which described the high MIC₅₀ of ATM-AVI for NDM-positive *E. coli* in our study, we discussed the finding of a cluster of NDM-5 *E. coli* ST617 and its potential clinical implications. We agreed with the reviewer that the description of resistant mechanism was irrelevant, so we deleted the sentence and related reference 32.

The revised paragraph was shown in line 249-254, underlined: "In addition, a cluster of four ATM-AVI non-susceptible NDM-5-positive *E. coli* ST617 was found. NDM-5-positive *E. coli* with reduced susceptibility to ATM-AVI had been discovered worldwide, and molecular typing showed diverse clonal background (33). More than ten sequence types were identified, including ST617 (33). The spread of these

ATM-AVI-resistant clones could be a concern in clinical practice.”

Comment 16. Figure 2 is redundant to the Tables and can be moved to supplementary data.

Reply: As suggested by the reviewer, we moved Figure 2 to supplementary data (Supplementary figure 1). The “Figure 2A” was changed to “Supplementary Figure 1A” (line 131), and the “Figure 2B” was changed to “Supplementary Figure 1B” (line 142). The “Figure 3A” and “Figure 3B” were changed to “Figure 2A” and “Figure 2B,” respectively (line 168 and 174).

Comment 17. Figure 3: it would be nice if some statements could be made about the potential reasons for clonality, i.e. transmission/outbreak? Otherwise, this may exaggerate the interpretation of the results presented.

Reply: Thank you for your suggestion. We analyzed the epidemiological correlation among the 21 cefiderocol or ATM-AVI non-susceptible *E. coli* and *E. cloacae complex* isolates shown in Figure 3. The data are presented in the newly added

Supplementary Table 1. We revised the result section to describe the findings:

- Line 168-171, underlined: “Two of the four ST617 *E. coli* isolates (M119 and M137) were from patients who had recently been discharged from two adjacent hospital wards within a two-month period (Supplementary Table 1),”
- Line 174-177, underlined: “Analysis of the epidemiologic data showed that three ST316 *E. cloacae complex* isolates (M89, M101, M124) were from patients who had been admitted to the hematology ward within the same year, but their admission periods did not overlap (Supplementary Table 1).”

The interpretation of these findings was added to the discussion section (line 229-231, underlined): “Although our investigation of the epidemiologic relationship

among the cefiderocol non-susceptible *E. coli* or *E. cloacae* complex isolates did not support a hospital outbreak, the increase in prevalence of cefiderocol non-susceptibility among *Enterobacteriaceae* via the dissemination of certain bacterial clones is worthy of potential concern.”

Comments of reviewer #3:

Reviewer #3: In this manuscript, the author evaluated in vitro activities of Cefiderocol and Aztreonam-avibactam against Metallo- β -lactamase-producing Enterobacteriaceae, and further investigated the impact of inoculum size. Overall, this manuscript is well organized and well written. It's interesting that the inoculum size of CPE substantially impacted the MICs of cefiderocol and aztreonam-avibactam, which can be addressed in the future research.

Reply: Thank you for your comment. We agree with the reviewer that further research is warranted to evaluate the clinical impact of the inoculum size of CPE on treatment outcome.

April 18, 2023

Prof. Jann-Tay Wang
National Taiwan University Hospital
Department of Internal Medicine
No. 7 Chung-Shan S rd, Taipei, Taiwan
Taipei, -- Not Applicable --
Taiwan

Re: Spectrum00569-23R1 (In Vitro Activities and Inoculum Effects of Cefiderocol and Aztreonam-avibactam against Metallo- β -lactamase-producing *Enterobacteriaceae*)

Dear Prof. Jann-Tay Wang:

Your manuscript has been accepted, and I am forwarding it to the ASM Journals Department for publication. You will be notified when your proofs are ready to be viewed.

Sincerely,

Pablo Power
Editor, Microbiology Spectrum
